# Incremental Learning of Vision-Language Models via Task Subspace Projection and Dynamic LoRA

## Abstract

Recent pre-trained vision-language models usually face a Multi-Domain Task-Incremental Learning (MTIL) scenario in practice, where a set of classes of multi-modal tasks arrive incrementally. Due to privacy concerns and memory constraints, MTIL with pre-trained models encounters forgetting of knowledge from old tasks, degradation of zero-shot transfer capability, and underfitting of new-task knowledge. To overcome these challenges, previous MTIL methods attempt to learn a discriminative cross-task identification (CTI) module and an effective new-task adaptation (NTA) module. However, current CTI modules suffer from severe task confusion between seen and unseen tasks, and NTA modules cannot adaptively balance the performance and parameter cost while incorporating task-specific knowledge. To alleviate the above dilemmas, we propose an effective and efficient TSP-DLoRA method for MTIL, which consists of Task Subspace Projection (TSP) and Dynamic Low Rank Adapter (DLoRA) modules. Specifically, our TSP module includes a task identifier classifier based on task-specific subspaces and a feature projection strategy that can determine the identifier associated with samples from both seen and unseen tasks. Our DLoRA improves the knowledge adaptation from new tasks by dynamically assigning Low Rank Adapter (LoRA) across transformer layers based on the task distributions. Experimental evaluations across 11 datasets, using three performance metrics, demonstrate the effectiveness of our proposed method.

## 1 Introduction

Deep neural networks have achieved remarkable performance in numerous multi-modal understanding applications. Traditional supervised learning methods in multi-modal learning require access to the entire dataset during the training phase, these models are no longer updated once training is completed Van de Ven & Tolias (2019). However, real-world multi-modal applications often encounter a dynamic data stream and need to learn a sequence of tasks continuously, which is referred to as the Multi-Domain Task-Incremental Learning (MTIL) benchmark. Due to privacy concerns or memory constraints, multi-modal models cannot access the previously seen tasks and suffer from severe catastrophic forgetting issue on MTIL benchmark.

With the powerful zero-shot capability of pre-trained multi-modal models (e.g., CLIP Radford et al. (2021)), existing approaches on MTIL benchmark consist of two modules Tang et al. (2024); Yu et al. (2024). 1) Cross-task identification (CTI) module: design a discriminative task identifier classifier to determine which task the sample belongs to, covering both seen and unseen tasks. 2) New-task adaptation (NTA) module: adapt the pre-trained model to different tasks by employing appropriate parameter-efficient fine-tuning (PEFT) methods or completely retraining all parameters of the model, as shown in Figure 1(a). Typically, the methods with pre-trained models on MTIL benchmark focus on zero-shot transfer capability preservation (especially on unseen tasks), old-task knowledge preservation, and new-task adaptation effectively and efficiently. However, we observe that the task confusion among seen and unseen tasks of existing CTI modules results from the degradation of the zero-shot transfer capability of the learned model. Moreover, prevalent NTA modules utilize fixed PEFT architecture for different tasks and cannot make a good trade-off between new-task performance and the task-specific parameter cost.

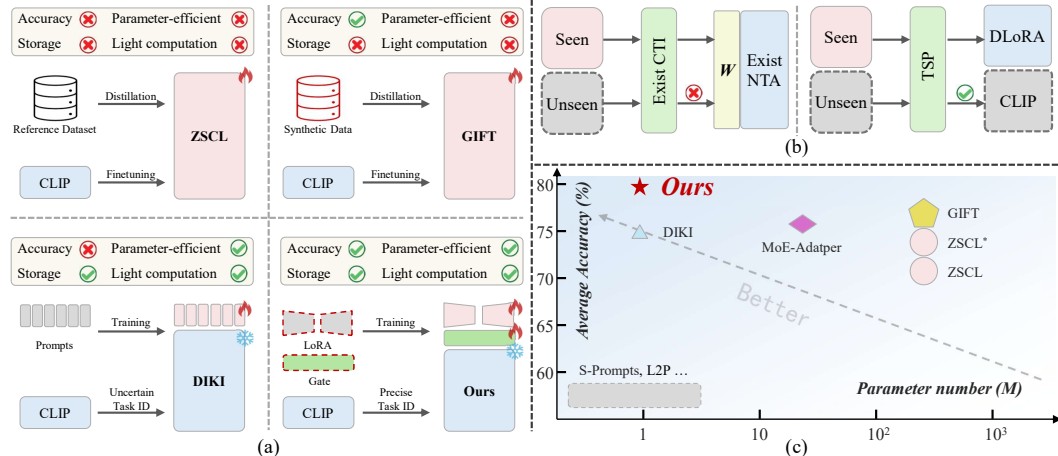

Figure 1: (a) Our method offers distinct advantages over existing methods. In comparison to methods ZSCL and GIFT, ours is more parameter-efficient, eliminating the need for additional storage to retain representative features. Compared with DIKI method, ours not only achieves higher accuracy but also demonstrates the ability to precisely determine whether samples belong to previously seen tasks. (b) Existing training-free CTI employs the same NTA operation on both seen and unseen tasks, and utilizes a weight "$W$" to restrict it. In contrast, our TSP module identifies the boundary of seen and unseen tasks, and employs original CLIP directly for samples from unseen tasks. (c) Compared to existing methods, ours achieves optimal performance in average accuracy (both seen and unseen tasks) and trainable parameters.

Motivated by the above observation, we propose the TSP-DLoRA method on MTIL benchmark. The CTI related module terms Task Subspace Projection (TSP) decomposes the features of each task into task subspaces and leverages an energy to derive the minimal subspace that captures the task's principal features. The task identifiers of test samples are determined by comparing the projections of corresponding features onto each seen tasks' subspaces. Additionally, The TSP module establishes a static threshold as the decision boundary to distinguish between seen and unseen tasks. As shown in Figure 1(b), for samples identified as from seen tasks, the corresponding task-specific module is employed for classification. Conversely, for samples classified as from unseen tasks, the zero-shot capability of the original CLIP model is utilized for classification. The NTA related module called Dynamic Low Rank Adapter (DLoRA) leverages Low Rank Adapter (LoRA) Hu et al. (2022) and incorporates a gating mechanism to dynamically determine whether to engage the LoRA module based on the complexity of the task distribution. By integrating the TSP and DLoRA modules, our method maintains high performance in both task identification and class classification while fine-tuning only a minimal number of parameters, as shown in Figure 1(c).

The contributions of this work are threefold: 1) We propose the TSP module, which accurately identifies sample identifiers by maintaining subspaces for seen tasks. It achieves over 93% accuracy across both seen and unseen tasks. 2) We propose the DLoRA module, which dynamically activates LoRA modules based on task distributions, enabling the model to adaptively learn from and perform inference on samples from different tasks. 3) Extensive experiments on benchmark datasets demonstrate that the TSP-DLoRA method achieves state-of-the-art (SOTA) results across all three evaluation metrics on the MTIL benchmark, while training only 0.86% of the parameters and requiring no additional storage.

## 2 RELATED WORKS

### 2.1 INCREMENTAL LEARNING

Incremental learning approaches can be classified into four categories: 1) Regularization-based incremental learning, which leverages regularization terms to guide the model's optimization process. Notable methods include EWC Kirkpatrick et al. (2017) and LwF Li & Hoiem (2017). 2) Rehearsal-

based incremental learning Li & Hoiem (2017); Rebuffi et al. (2017); Wu et al. (2019); Hou et al. (2018); Lee et al. (2019); Hou et al. (2019); Park et al. (2021). These methods aim to preserve knowledge by retaining or generating representative samples or features from seen tasks, which are then trained together with data from unseen tasks. Prominent works include iCaRL Rebuffi et al. (2017), ZSCL Zheng et al. (2023), and GIFT Wu et al. (2025). 3) Network expansion-based incremental learning Ostapenko et al. (2019); Yoon et al. (2017); Xu & Zhu (2018); Li et al. (2019). This approach accommodates new tasks by dynamically expanding the model architecture. The representative method is DEN Yoon et al. (2017). 4) Incremental learning via parameter-efficient fine-tuning (PEFT) Jung et al. (2023); Tang et al. (2023); Zhou et al. (2025); Chen et al. (2024); Gao et al. (2023). Leveraging the robust zero-shot transfer capability of pre-trained models, this category has emerged as a prevalent strategy in incremental learning. These methods typically freeze the backbone of pre-trained models and fine-tune a small subset of parameters using techniques such as LoRA Meral et al. (2024), Adapters Gao et al. (2024), or Prompt Wang et al. (2022c). The well-known approaches to incremental learning via PEFT include L2P Wang et al. (2022c), DualPrompt Wang et al. (2022b), S-Prompt Wang et al. (2022a), MoE-Adapter Yu et al. (2024), and DIKI Tang et al. (2024). Unlike existing approaches that rely on a fixed structure, our method introduces a novel framework by dynamically adjusting the PEFT structure based on the input, which achieves superior performance compared to all traditional PEFT techniques in incremental learning.

## 2.2 MULTI-DOMAIN TASK-INCREMENTAL LEARNING

The multi-domain task-incremental learning (MTIL) benchmark is first introduced in the work Zheng et al. (2023). This work proposes the ZSCL method, which leverages knowledge distillation, utilizing a reference dataset to transfer knowledge from the old models to the new one. A related method , GIFT Wu et al. (2025), adopts a similar strategy by generating representative features to substitute for the reference dataset. Nevertheless, both techniques rely on full fine-tuning, resulting in significant computational cost. Existing PEFT related methods in MTIL include MoE-Adapter Yu et al. (2024) and DIKI Tang et al. (2024). The MoE-Adapter approach, while training a quarter of its parameters, still incurs considerable computational cost when applied to parameter-heavy models like CLIP. DIKI trains a model with fewer parameters; however, its static fine-tuning architecture struggles to accommodate tasks with pronounced distributional disparities, leading to diminished performance. In contrast, our proposed method employs a dynamic fine-tuning structure with only 0.86% of total trainable parameters, while adapting to varying task distributions and offering an efficient and effective solution for the MTIL benchmark.

## 2.3 DISCRIMINATIVE CROSS-TASK IDENTIFICATION

On the MTIL benchmark, the model is required to determine the task identifier of a test sample initially during inference, and infer the specific class based on the identifier. This process requires a highly effective Cross-Task Identification (CTI) module. Currently, two CTI modules, i.e., DDAS module in MoE-Adapter and DAIC module in DIKI are proposed to improve the efficacy of task identification. Specifically, DDAS involves maintaining a distinct linear classifier for each new task, optimized according to its specific distribution before training the model. During inference, task identifiers for test samples are predicted using the linear classifiers. However, DDAS module introduces a substantial number of learnable parameters, resulting in significant computational overhead during both training and inference. Moreover, it requires task-specific adjustments to the classifier hyperparameters. DAIC avoids the introduction of additional trainable parameters; instead, it stores the mean and variance of each new task's distribution. During inference, it models each seen task as a Gaussian distribution and computes the similarity between the test sample and these distributions. Existing CTI modules encounter the task confusion among seen and unseen tasks. To achieve an effective CTI module, we propose a TSP method to learn a distinct subspace for each task.

## 3 APPROACH

### 3.1 PRELIMINARIES

**Benchmark.** Consider a pre-trained VLM that undergoes incremental learning through a sequence of tasks, which originate from $\mathcal{T}$ distinct domains, denoted as $\mathcal{D} = \{D_1, D_2, \ldots, D_{\mathcal{T}}\}$. Each

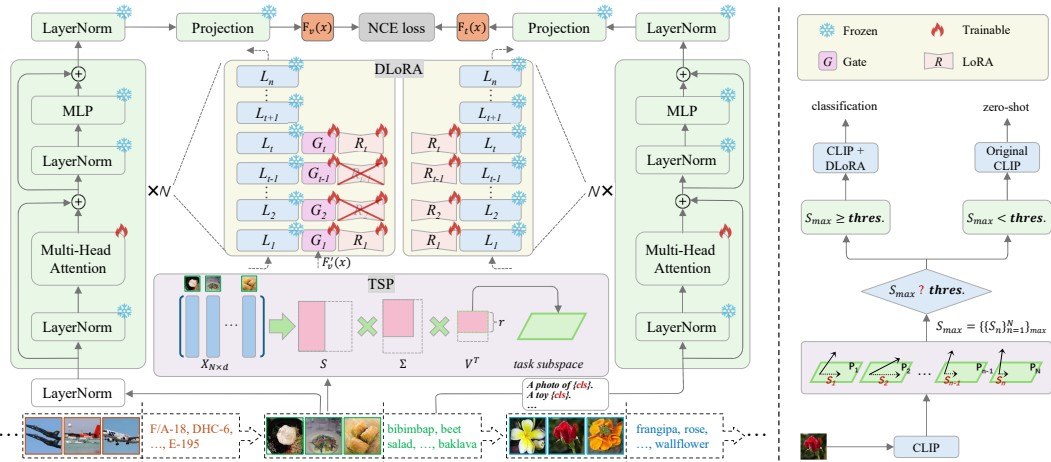

Figure 2: Left: The training process of our method. The TSP module decomposes image features by SVD technology and selects top $r$ ranks from the right singular vector matrix to be the task subspace. DLoRA module integrates LoRA into the first $L_t$ transformer layers of both the image and text encoders. The gating mechanism before the LoRA modules in the image encoder determines whether to activate the LoRA based on the feature $F'_v(x)$ derived from the original CLIP. Only the LoRA and the gating mechanism modules are trained, while the remaining parts are kept frozen. Right: Inference period. We compute the angle between the raw feature extracted from original CLIP and the subspaces associated with each seen task. The resulting similarity is compared against a threshold, denoted as "thres.". The sample is classified using the corresponding DLoRA module if the similarity exceeds the "thres.". Otherwise, classification relies on the original CLIP model.

domain $D_t$ comprises $N$ samples, represented as $(x_n^t, y_n^t)_{n=1}^{N_t}$, where $x_n^t$ denotes a raw image and $y_n^t$ represents its corresponding one-hot encoded ground truth label. There also exists an associated class set, defined as $C_t = \{c_i^t\}_{i=1}^{M_t}$, where each $c_i^t$ is a textual label describing a specific class, and $M_t$ is the label space size of task $t$. On the MTIL benchmark, access to the data of domain $D_t$ is restricted exclusively to the $t$-th phase of incremental learning. Furthermore, the class sets across domains are pairwise disjoint, such that $C_i \cap C_j = \emptyset$ for all $i \neq j$, ensuring that each domain possesses a unique collection of classes. Additionally, the data distributions differ across domains, expressed as $\mathbb{P}_i \neq \mathbb{P}_j$ for $i \neq j$, where $\mathbb{P}_i$ signifies the data distribution of domain $D_i$. During the inference phase, the model requires performing inference in a specific label space (e.g., $C_t$). Consequently, obtaining an accurate task identifier $t$ for each test sample is a crucial aspect of the MTIL task.

**CLIP Models.** Pre-trained VLMs (e.g., CLIP Radford et al. (2021)) typically comprise two encoders: image encoder $F_v$ and text encoder $F_t$. These pre-trained VLMs consistently perform a preprocessing step that converts the class name $c_i^t$ into a sentence using a set of predefined templates, such as "{a photo of {$c_i^t$}.}". This sentence is subsequently encoded into a text embedding $t_i$ by the tokenizer. CLIP models are trained by contrastive loss Park et al. (2020), where the optimize objective can be denoted as:

$$L = -\sum_{i=1}^{N_t} \log \left( \frac{\exp\left(\text{sim}\left(F_v(x_i), F_t(t_i)\right)/\tau_c\right)}{\sum_{j=1}^{N_t} \exp\left(\text{sim}\left(F_v(x_i), F_t(t_j)\right)/\tau_c\right)} \right) \quad , \tag{1}$$

$F_v(x_i), F_t(t_i)$ are the features extracted by the visual and text encoders, $\tau_c$ represents the temperature, and $\text{sim}(u, v) = \frac{u^T \cdot v}{\|u\| \|v\|}$ is the cosine similarity function. The contrastive loss facilitates the CLIP model in capturing the inter-modal similarity between the image and text embeddings.

### 3.2 FRAMEWORK OVERVIEW

In this work, we propose a parameter-efficient framework aimed at enhancing the incremental learning capability of CLIP models from two key perspectives. First, to facilitate the acquisition of new

tasks, we introduce the Dynamic LoRA (DLoRA). It dynamically adapts the fine-tuning modules, which enables the model to learn new tasks with a minimal number of trainable parameters and effectively accommodate a diverse range of tasks. Second, to preserve the zero-shot transfer capability of the pre-trained model, we develop the Task Subspace Projection (TSP) module, a newly designed CTI module that leverages projection on task subspace to determine the identifier of each sample.

## 3.3 DYNAMIC LoRA

**New knowledge injection strategy via LoRA.** When the pre-trained model is trained on new tasks, for the image encoder $F_v$ and the text encoder $F_t$ of the CLIP model, we assume that each encoder comprises $L_n$ transformer layers. The LoRA module is applied to the first $t$ layers, which can be denoted as $R_t, (t \leq n)$, as depicted in Figure 2. Specifically, for the weight matrix $W$ of a linear layer, we decompose it into the product of two smaller matrices:

$$\triangle W = W_{down} W_{up} \quad , \tag{2}$$

where $W_{down} \in \mathbb{R}^{d \times r}$ and $W_{up} \in \mathbb{R}^{r \times d}$. In the self-attention mechanism of the first $L_t$ layers, we follow Liang & Li (2024) and incorporate LoRA into the key and value, which are updated according to the following operations:

$$
\begin{aligned}
K_r &= (W_k + \mathbf{e} * \triangle W_k) K_{init} + b_k \\
V_r &= (W_v + \mathbf{e} * \triangle W_v) V_{init} + b_v
\end{aligned} \quad , \tag{3}
$$

$\mathbf{e}$ represents a scaling factor, $W_k$, $W_v$, $K_{init}$, $V_{init}$, $b$ are the initial weight, key, value, bias of transformer layers. We employ LoRA for both the visual and text encoders, while keeping the model's backbone parameters entirely frozen.

**Is injecting LoRA to all top $L_t$ layers always the optimal choice?** Conventional LoRA-based methods typically involve injecting learnable modules at predetermined fixed positions, relying on the assumption that training datasets are consistently drawn from the same distribution. However, on the MTIL benchmark, models must dynamically adapt to datasets exhibiting diverse distributions Tang et al. (2024). Moreover, these datasets also vary in terms of data volume and number of classes. Employing a static learning strategy across such heterogeneous datasets may result in overfitting to simpler datasets or underfitting to more complex ones. In this work, we observe this challenge and propose a dynamic LoRA injection strategy to deal with the unique properties of each dataset. Specifically, we enhance the capability of LoRA in the top $L_t$ transformer layers by introducing a Gumbel-based gating mechanism, which dynamically determines whether to inject LoRA to the corresponding layer based on the input feature, as shown in the left of Figure 2.

**Gumbel-based gating mechanism.** During the training phase, to avoid the feature space instability arising from parameter optimization, we utilize the feature outputs of the original, frozen, pre-trained CLIP model $F_v'(x)$ as inputs to the Gumbel-based gating mechanism. We employ a linear layer $H$ in the transformer layer, which maps the original image features $F_v'(x)$ to a $K$-dimensional feature space, as well as a Gumbel distribution which is used to generate the samples uniformly. The overall Gumbel logit for every sample can be denoted as:

$$G_i = \frac{\exp(\log\left(H(F_v'(x)) + u_i\right)/\tau_g)}{\sum_{j=1}^{K} \exp(\log\left(H(F_v'(x)) + u_j\right)/\tau_g)} \quad , \tag{4}$$

$u_i = -log(-log(U_i))$ is randomly sampled from a normal distribution, where $U_i \sim U(0,1)$. $\tau_g$ represents the temperature. Our gating mechanism operates with only two values, 1 and 0. 1 denotes injecting LoRA at this layer and 0 represents not. To facilitate this binary decision process, we set $K = 2$ to generate hard-coded representations that guide the LoRA injection strategy as follows:

$$
\begin{aligned}
K_r' &= G_1 K_{init} + G_2 K_r \\
V_r' &= G_1 V_{init} + G_2 V_r
\end{aligned} \quad , \tag{5}
$$

By leveraging features from a stable space to guide its gating mechanism, our proposed DLoRA module dynamically tailors its LoRA injection strategy to each sample based on the complexity of the task distribution.

## 3.4 TASK SUBSPACE PROJECTION

**Why and what is TSP?** On the MTIL benchmark, the model incrementally learns new tasks and performs inference across all tasks. Because of the significant differences between distributions of these tasks, the performance of the CTI module is pivotal to the overall effectiveness of the model. Current approaches typically adopt two strategies: 1) Train an additional classifier to identify the task identifier Yu et al. (2024), which introduces extra trainable parameters and elevates training costs. 2) Identify all samples as from seen tasks directly but apply weights to restrict the use of task-specific modules Tang et al. (2024); however, this introduces additional uncertainty for samples from both seen and unseen tasks. To overcome these challenges, we introduce Task Subspace Projection (TSP), a novel training-free CTI module that leverages singular value decomposition (SVD) to extract each seen task's subspace and differentiates which task a test sample belongs to by the subspaces. For samples identified as from seen tasks, the model employs the corresponding task-specific module for classification. Conversely, for samples identified as from unseen tasks, the model relies entirely on the untrained original CLIP model for inference, fully utilizing the zero-shot capability of the pre-trained model.

**Construction of task-specific subspaces.** When acquiring a new task, the TSP module first extracts features from all samples of the new task using the original CLIP model, $X = [F_v'(x_1), F_v'(x_2), \ldots, F_v'(x_N)] \in \mathbb{R}^{N \times d}$, where $N$ is the total number of samples from new task and $d$ is the feature dimension. As shown in Figure 2, these features are then subjected to SVD to extract the subspace associated with the new task:

$$X = U\Sigma V^T, \tag{6}$$

where $U \in \mathbb{R}^{N \times N}$ is the left singular vector matrix, $\Sigma \in \mathbb{R}^{N \times d}$ is the singular values matrix and $V \in \mathbb{R}^{d \times d}$ is the right singular vector matrix.

The value of rank $r$ during SVD directly determines the dimension of the task subspace, which is critical to the TSP module. To maximize principal component retention within the task subspace while minimizing computational costs, we propose an adaptive energy-based selection strategy to ensure consistent representation across diverse tasks. The diagonal elements in singular values matrix correspond to the singular value vectors, thus we determine the number of ranks $r$ by analyzing the energy proportion of each singular value vector. Specifically, we calculate the variance of the singular value matrix and then compute the cumulative sum of the energy proportions for the top $k$ ranks:

$$\mathbb{E}_k = \frac{\sum_{i=1}^{k} \sigma_i^2}{\sum_{i=1}^{q} \sigma_i^2} \quad, \tag{7}$$

where $\sigma_i$ is the $i$-th diagonal element of the matrix, and $q = min(N, d)$ is the smaller of the number of image features $N$ and the feature dimension $d$. We then select the smallest $k$ such that the cumulative energy of the first $k$ singular values reaches or exceeds the preset threshold energy:

$$k = min\{k | \mathbb{E}_k \geq energy\} \quad, \tag{8}$$

where $energy$ is a hyperparameter. We set the final selected $r$ to $k + 1$ to ensure that the chosen rank exists.

Our aim is to ensure that the distribution of the task-specific subspace maximally reflects the distribution of the new task. To this end, we select the first $r$ rows of the right singular value matrix to represent the feature distribution of the task, denoted as $V_r = V_{[:,0:r]}, V_r \in \mathbb{R}^{d \times r}$. To facilitate efficient computation during inference, we store the orthogonal projection operator of this subspace:

$$\mathcal{P} = V_r V_r^T \quad, \tag{9}$$

where $\mathcal{P} \in \mathbb{R}^{d \times d}$. We store a $\mathcal{P}$ for every seen task. Therefore, during the $t$-th incremental learning period, $\{\mathcal{P}_1, \mathcal{P}_2, \ldots, \mathcal{P}_t\}$ are available.

**Inference.** In the inference phase, for a test image $x$ with unknown task identifier, we extract $F_v'(x)$ using the original CLIP model, ensuring consistency with the task subspaces. For each seen task, we compute the projection of $F_v'(x)$ onto its corresponding specific task subspace as follows:

$$\widetilde{F_v'(x)}_t = \mathcal{P}_t F_v'(x) \quad, \tag{10}$$

Subsequently, we calculate the angle between $F'_v(x)$ and $\widetilde{F'_v(x)}$ using cosine similarity as follows:

$$S_t = \frac{F'_v(x)\widetilde{F'_v(x)}_t}{||F'_v(x)|| \cdot ||\widetilde{F'_v(x)}_t||} \quad , \tag{11}$$

We focus exclusively on the vector that forms the smallest angle with the feature subspace of the seen tasks, which corresponds to the maximum value in $S_t$. To intuitively determine whether a given test sample belongs to a seen or unseen task, we define a threshold, denoted as "$Thres.$". As shown in Figure 2, by comparing the maximum value in $S_t$ with "$Thres.$", the task identifier of the test sample is derived:

$$task\ id = \begin{cases} \arg\max_{i \in \{1,2,...,t\}} S_i & \text{for } S_{max} \geq Thres. \\ -1 & \text{for } S_{max} < Thres. \end{cases} , \tag{12}$$

where $S_{max} = \max\{S_1, S_2, ..., S_t\}$, and $-1$ represents the test sample belongs to unseen tasks.

The TSP module accurately assigns a task identifier to each sample during the inference phase. If the task identifier is classified as seen tasks, the model applies the corresponding DLoRA module to infer the specific label. Otherwise, the model directly employs the original CLIP model, leveraging its robust pre-trained knowledge to determine the label.

## 4 EXPERIMENTS

### 4.1 EXPERIMENTAL SETTING

**Dataset and metrics.** We follow Zheng et al. (2023) and evaluate our method on the MTIL benchmark, which comprised 11 datasets: Aircraft Maji et al. (2013), Caltech101 Fei-Fei et al. (2004), CIFAR100 Krizhevsky et al. (2009), DTD Cimpoi et al. (2014), EuroSAT Helber et al. (2019), Flowers Nilsback & Zisserman (2008), Food Bossard et al. (2014), MNIST Deng (2012), Oxford-Pet Parkhi et al. (2012), StanfordCars Krause et al. (2013) and SUN397 Xiao et al. (2010), with a total of 1201 classes across distinct distributions. The model's performance is assessed using three primary metrics: "Transfer", "Last", and "Avg". Further details regarding both the datasets and theses evaluation metrics are provided in the supplementary materials.

**Comparison methods.** We compare our method with two categories of SOTA methods, which are full parameter fine-tuning (FPFT) and PEFT methods. FPFT methods leverage rehearsal-based techniques and knowledge distillation to retain the old knowledge, necessitating updating all parameters and external storage during training. The comparison methods in our experiments include Continual-FT, iCaRL, LwF-VR Ding et al. (2022), WiSE-FT Wortsman et al. (2022), ZSCL and GIFT. PEFT methods learn new tasks by updating only a small set of trainable parameters. Such methods include L2P, DualPrompt, S-Prompt, MoE-Adapter and DIKI. Our proposed method falls within this category.

**Implementation details.** As in Zheng et al. (2023), we utilize CLIP ViT-B/16 as our backbone for all the experiments. We apply our DLoRA module to the first 8 transformer layers of both visual and text encoders and fix the rank at 4. For the gating mechanism, we employ a learning rate of 2.0 and set the temperature to 1.0. Additionally, we conduct an ablation study on the learning rate and temperature of the gating mechanism, details are provided in the supplementary materials. Both the DLoRA and gating mechanism modules adopt stochastic gradient descent (SGD) as the optimizer, coupled with cosine annealing to adjust the learning rate. For the TSP module, we establish a static energy level of 95% across all 11 tasks to dynamically determine the rank. The threshold is set to 0.96 to serve as the decision boundary between seen and unseen tasks. The model is trained for 10 epochs on each task using an NVIDIA 4090 GPU.

### 4.2 EXPERIMENTAL RESULTS

The main experimental results are presented in Table 1. "Extra." denotes whether external data is required during the training process. "Param." refers to the total number of trainable parameters. "Zero-shot" represents the inference performance using only the pre-trained knowledge, serving as the lower bound of the current benchmark. "Full Fine-tune" involves fully fine-tuning the CLIP

Table 1: Comparison with SOTA on MTIL benchmark in terms of "Transfer", "Average", and "Last" metrics (%). "Ours" denotes our method. The presented results are derived from the Order-I, for Order-II results, please refer to the supplemental materials.

| | Method | Extra. | Param. | Aircraft | Caltech101 | CIFAR100 | DTD | EuroSAT | Flowers | Food | MNIST | OxfordPet | StanfordCars | SUN397 | *Average* |
|---|---|---|---|---|---|---|---|---|---|---|---|---|---|---|---|
| **CLIP** | Zero-shot | ✗ | - | 24.3 | 88.4 | 68.2 | 44.6 | 54.9 | 71.0 | 88.5 | 59.4 | 89.0 | 64.7 | 65.2 | 65.3 |
| | Full Fine-tune | ✗ | 211M | 62.0 | 95.1 | 89.6 | 79.5 | 98.9 | 97.5 | 92.7 | 99.6 | 94.7 | 89.6 | 81.8 | 89.2 |
| **Transfer** | Continual-FT | ✓ | 211M | - | 67.1 | 46.0 | 32.1 | 35.6 | 35.0 | 57.7 | 44.1 | 60.8 | 20.5 | 46.6 | 44.6 |
| | iCaRL | ✓ | 211M | - | 56.6 | 44.6 | 32.7 | 39.3 | 46.6 | 68.0 | 46.0 | 77.4 | 31.9 | 60.5 | 50.4 |
| | LwF-VR | ✓ | 211M | - | 77.1 | 61.0 | 40.5 | 45.3 | 54.4 | 74.6 | 47.9 | 76.7 | 36.3 | 58.6 | 57.2 |
| | WiSE-FT | ✓ | 211M | - | 73.5 | 55.6 | 35.6 | 41.5 | 47.0 | 68.3 | 53.9 | 69.3 | 26.8 | 51.9 | 52.3 |
| | ZSCL | ✓ | 211M | - | 86.0 | 67.4 | 45.4 | 50.4 | 69.1 | 87.6 | 61.8 | 86.8 | 60.1 | 66.8 | 68.1 |
| | GIFT | ✗ | 211M | - | 88.5 | 69.8 | 46.0 | 49.4 | 68.5 | 87.1 | 69.9 | 88.9 | 57.7 | 67.7 | 69.3 |
| | L2P | ✗ | 0.5M | - | 65.6 | 50.9 | 41.4 | 49.3 | 71.8 | | 36.3 | 77.5 | 55.3 | 53.4 | 53.2 |
| | DualPrompt | ✗ | 1.8M | - | 56.7 | 51.4 | 28.7 | 33.7 | 45.6 | 70.9 | 59.5 | 77.7 | 49.5 | 50.4 | 52.4 |
| | S-Prompts | ✗ | 0.5M | - | 67.3 | 49.4 | 26.7 | 39.7 | 47.1 | 70.2 | 34.3 | 78.9 | 56.7 | 52.2 | 52.2 |
| | MoE-Adapter | ✓ | 59.8M | - | 87.9 | 68.2 | 44.4 | 49.9 | 70.7 | 88.7 | 59.7 | 89.1 | 64.5 | 65.5 | 68.9 |
| | DIKI | ✗ | 1.8M | - | 92.9 | 69.1 | 43.2 | 43.9 | 65.4 | 85.3 | 56.0 | 88.4 | 65.4 | 65.6 | 67.4 |
| | **Ours** | ✗ | 1.8M | - | 93.5 | 68.5 | 43.5 | 48.5 | 70.8 | 86.1 | 64.7 | 89.1 | 66.4 | 62.6 | **69.4** |
| **Average** | Continual-FT | ✓ | 211M | 25.5 | 81.5 | 59.1 | 53.2 | 64.7 | 51.8 | 63.2 | 64.3 | 69.7 | 31.8 | 49.7 | 55.9 |
| | iCaRL | ✓ | 211M | 35.5 | 89.2 | 72.2 | 60.6 | 68.8 | 70.0 | 78.2 | 62.3 | 81.8 | 41.2 | 62.5 | 65.7 |
| | LwF-VR | ✓ | 211M | 29.6 | 87.7 | 74.4 | 59.5 | 72.4 | 63.6 | 77.0 | 66.7 | 81.2 | 43.7 | 60.7 | 65.1 |
| | WiSE-FT | ✓ | 211M | 26.7 | 86.5 | 64.3 | 57.1 | 65.7 | 58.7 | 71.1 | 70.5 | 75.8 | 36.9 | 54.6 | 60.7 |
| | ZSCL | ✓ | 211M | 45.1 | 92.0 | 80.1 | 64.3 | 79.5 | 81.6 | 89.6 | 75.2 | 88.9 | 64.7 | 68.0 | 75.4 |
| | GIFT | ✗ | 211M | 51.9 | 93.9 | 81.4 | 67.7 | 80.3 | 82.8 | 89.3 | 80.6 | 90.3 | 63.1 | 68.9 | 77.3 |
| | L2P | ✗ | 0.5M | 38.0 | 85.2 | 78.2 | 61.3 | 72.9 | 74.9 | 79.7 | 59.1 | 82.0 | 59.7 | 55.4 | 67.9 |
| | DualPrompt | ✗ | 1.8M | 37.8 | 84.3 | 78.6 | 60.1 | 71.1 | 73.2 | 79.1 | 73.9 | 82.3 | 55.1 | 52.8 | 68.0 |
| | S-Prompts | ✗ | 0.5M | 37.5 | 92.5 | 77.5 | 58.2 | 76.4 | 74.1 | 78.8 | 57.9 | 83.0 | 60.8 | 54.4 | 68.3 |
| | MoE-Adapter | ✓ | 59.8M | 50.2 | 91.9 | 83.1 | 69.4 | 78.9 | 84.0 | 89.1 | 73.7 | 89.3 | 67.7 | 66.9 | 76.7 |
| | DIKI | ✗ | 1.8M | 45.4 | 95.7 | 83.0 | 65.0 | 78.2 | 82.5 | 87.1 | 71.7 | 90.0 | 67.2 | 66.6 | 75.7 |
| | **Ours** | ✗ | 1.8M | 50.4 | 96.3 | 83.3 | 67.5 | 80.2 | 85.7 | 87.5 | 77.3 | 90.8 | 69.6 | 63.9 | **77.5** |
| **Last** | Continual-FT | ✓ | 211M | 31.0 | 89.3 | 65.8 | 67.3 | 88.9 | 71.1 | 85.6 | 99.6 | 92.9 | 77.3 | 81.1 | 77.3 |
| | iCaRL | ✓ | 211M | 35.8 | 93.0 | 77.0 | 70.2 | 83.3 | 88.5 | 90.4 | 86.7 | 93.2 | 81.2 | 81.9 | 80.1 |
| | LwF-VR | ✓ | 211M | 20.5 | 89.8 | 72.3 | 67.6 | 85.5 | 73.8 | 85.7 | 99.6 | 93.1 | 73.3 | 80.9 | 76.6 |
| | WiSE-FT | ✓ | 211M | 27.2 | 90.8 | 68.0 | 68.9 | 86.9 | 74.0 | 87.6 | 99.6 | 92.6 | 77.8 | 81.3 | 77.7 |
| | ZSCL | ✓ | 211M | 40.6 | 92.2 | 81.3 | 70.5 | 94.8 | 90.5 | 91.9 | 98.7 | 93.9 | 85.3 | 80.2 | 83.6 |
| | GIFT | ✗ | 211M | 47.9 | 95.6 | 82.8 | 75.1 | 97.3 | 94.2 | 91.7 | 99.2 | 94.2 | 87.0 | 80.9 | 86.0 |
| | L2P | ✗ | 0.5M | 38.0 | 87.1 | 84.2 | 72.9 | 86.0 | 96.1 | 89.2 | 99.0 | 94.1 | 79.6 | 76.0 | 82.0 |
| | DualPrompt | ✗ | 1.8M | 37.8 | 87.1 | 84.6 | 71.8 | 89.2 | 96.3 | 89.1 | 99.1 | 94.5 | 79.9 | 76.5 | 82.3 |
| | S-Prompts | ✗ | 0.5M | 37.5 | 95.1 | 83.7 | 70.2 | 97.5 | 96.5 | 89.0 | 99.1 | 94.0 | 79.5 | 75.8 | 83.4 |
| | MoE-Adapter | ✓ | 59.8M | 49.8 | 92.2 | 86.1 | 78.1 | 95.7 | 94.3 | 89.5 | 98.1 | 89.9 | 81.6 | 80.0 | 85.0 |
| | DIKI | ✗ | 1.8M | 45.4 | 95.9 | 86.0 | 73.0 | 97.8 | 96.8 | 89.3 | 99.3 | 94.4 | 81.8 | 76.4 | 85.1 |
| | **Ours** | ✗ | 1.8M | 50.4 | 96.6 | 86.7 | 76.5 | 98.3 | 98.2 | 89.3 | 99.6 | 94.6 | 84.3 | 77.1 | **86.5** |

model with 11 tasks, establishing the upper bound of performance. Among all the methods, our proposed method, which integrates the DLoRA and TSP modules, achieves SOTA performance on all the average of "Transfer", "Average", and "Last" metrics. The method most comparable to ours is GIFT. However, our approach requires only 0.86% of the training parameters used by GIFT, while achieving comparable or even superior performance across all three metrics. Furthermore, our approach eliminates the need to store additional representative samples from previous seen tasks.

We also follow Tang et al. (2024) and evaluate our method on the Order-II and 16-shot MTIL-FS benchmark. Our method achieves optimal results compared to the baseline. Details are provided in the supplementary materials.

## 4.3 ANALYSIS

**Effect of TSP module.** To assess the effectiveness of the TSP module, we replace the task identifier classifier in DIKI with the TSP module while keeping the fine-tuning strategy unchanged. The results are demonstrated in Table 2. Asterisk (*) denotes the experimental results obtained from our experiments, which may differ from the original paper Tang et al. (2024) due to variations in implementation or experimental conditions. The TSP module improves performance across all the "Transfer", "Average", and "Last" metrics. This suggests that TSP not only enhances task identification accuracy but can also serve as a plug-and-play component for various methods.

Table 2: The ablation experiments for DLoRA and TSP modules of our proposed method. Asterisk (*) denotes the practical results obtained from our experiments.

| Method | Trans. | Avg. | Lst. |
|---|---|---|---|
| DIKI* | 67.4 | 75.7 | 85.1 |
| DIKI*+TSP | 68.9 | 76.3 | 85.3 |
| LoRA+TSP | 69.3 | 76.5 | 85.4 |
| DLoRA+TSP | 69.4 | 77.5 | 86.5 |

Table 3: Computational cost of three PEFT methods. "GPU." denotes the GPU memory requirement, "Train." and "Infer" are the training and inference speeds.

| Method | Time. | GPU. | Train. | Infer. |
|---|---|---|---|---|
| MoE-Adapter | 5.3h | 48 GB | 1.58s/it | 0.63s/it |
| DIKI | 2.6h | 24 GB | 0.36s/it | 0.33s/it |
| Ours | 2.4h | 24 GB | 0.38s/it | 0.23s/it |

**Effect of DLoRA module.** To investigate the effectiveness of the dynamic gating mechanism in DLoRA, we conduct experiments combining TSP with standard LoRA. As shown in Table 2, our proposed DLoRA module outperforms the baseline of using LoRA alone across all three metrics, with notable improvements exceeding 1.0% and 1.1% on the "Average" and "Last" metrics respectively. These results align with our expectations, as DLoRA is designed to enhance the capability of learning new tasks.

**Visualization of TSP module.** Figure 3 illustrates the similarity distributions between the features of 500 random test samples and their projections onto each seen tasks' subspace, with an energy of 0.95. A threshold, represented by the red line in the figure, is set at 0.96. Most of the test samples (93% in our experiments) exhibit similarities that exceed the threshold, indicating that they are correctly classified for their corresponding tasks, with median similarity values around 0.97. The height of the boxes indicates that the similarity distributions of the test samples are highly concentrated; this demonstrates that our TSP module can efficiently extract critical task-specific

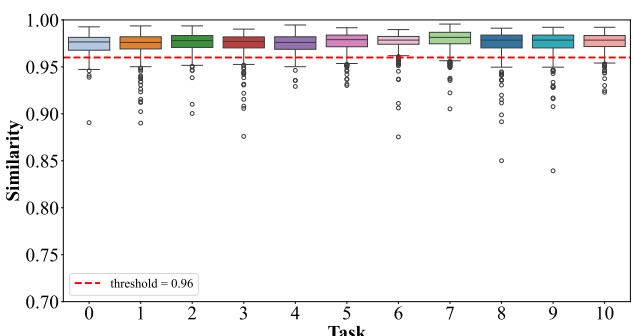

Figure 3: The distribution of similarity scores for the TSP module across different tasks. The red line denotes a threshold set at 0.96. Most of the samples exhibit similarity scores above this threshold. The dense similarity scores distribution also reveals the TSP module's capability to effectively integrate and extract critical information from each task.

information. We also present the distribution for each individual task and the influence of energy and threshold values on the TSP module in the supplementary materials, which shows that the identifiers for most samples are correctly assigned, with only a small fraction misclassified as "unseen" and an even smaller number incorrectly assigned to other tasks.

**Computational Cost.** We evaluate our method against two representative PEFT methods, MoE-Adapter and DIKI, on the MTIL benchmark. As shown in Table 3, our method consistently outperforms MoE-Adapter across three key metrics, including total time, GPU memory, and inference speed. Notably, our method significantly boosts the inference speed, this is due to the TSP module, which enables the model to leverage the native zero-shot capability of the original CLIP model for a subset of samples.

## 5 CONCLUSION

In this work, we introduce TSP-DLoRA, a parameter-efficient method composed of two key modules. The TSP module, operates as a training-free discriminative CTI module, accurately identifying task identifiers for samples from both seen and unseen tasks, effectively preserving the zero-shot transfer capabilities of pre-trained models. The NTA module termed DLoRA leverages a gating mechanism to dynamically determine the activation of the LoRA module based on the task distribution, thereby facilitating the model's ability to adapt to new tasks. Extensive experimental results demonstrate that both modules perform well independently but also, when integrated, surpass all existing methods at a remarkably low training cost.

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

## A  APPENDIX

## B  EXPERIMENTAL DETAILS

**Experimental settings.** All experimental results are derived utilizing PyTorch Paszke et al. (2019). The batch size is set to 128 during the training phase, and set to 256 for the inference. To reduce the computational burden associated with both training and inference, experiments are performed with FP16 precision. In the context of our proposed DLoRA module, a perturbation of $1 \times 10^{-6}$ is applied to all values sampled via Gumbel sampling to address potential numerical instability. Both the DLoRA and gating mechanism modules adopt stochastic gradient descent (SGD) as the optimizer, coupled with cosine annealing to adjust the learning rate.

Table 4: Detailed information of 11 datasets.

| Dataset | Classes | Train | Test | Recognition Task |
|---------|---------|-------|------|------------------|
| Aircraft Maji et al. (2013) | 100 | 3334 | 3333 | aircraft series |
| Caltech101 Fei-Fei et al. (2004) | 101 | 6212 | 2465 | real-life object |
| CIFAR100 Krizhevsky et al. (2009) | 100 | 50000 | 10000 | real-life object |
| DTD Cimpoi et al. (2014) | 47 | 2068 | 1692 | texture recognition |
| EuroSAT Helber et al. (2019) | 10 | 18800 | 8100 | satellite location |
| Flowers Nilsback & Zisserman (2008) | 102 | 4706 | 2463 | flower species |
| Food Bossard et al. (2014) | 101 | 70700 | 30300 | food type |
| MNIST Deng (2012) | 10 | 60000 | 10000 | digital number |
| OxfordPet Parkhi et al. (2012) | 37 | 3680 | 3669 | animal species |
| StanfordCars Krause et al. (2013) | 196 | 8144 | 8041 | car series |
| SUN397 Xiao et al. (2010) | 397 | 88904 | 19850 | scene category |
| Total | 1201 | 316548 | 99913 | |

**Details of datasets.** We utilize the same datasets as Tang et al. (2024) to validate our approach. The detailed information for all datasets are demonstrated in Table 4.

**Metrics.** The "Transfer" metric focuses on assessing the forgetting of the model's zero-shot transfer capability, known as forward forgetting Tang et al. (2024), for task $i$, it is computed as the average performance over unseen tasks $i+1, i+2, \ldots, \mathcal{T}$. The "Last" metric measures the model's ability to learn new tasks while mitigating catastrophic forgetting of seen tasks, which corresponds to backward forgetting, for task $i$, it is determined by averaging the performance across the seen tasks $i, i-1, \ldots, 1$. The "Avg" metric considers both forward forgetting and backward forgetting. At each incremental learning step, it is computed as the average performance across all tasks $\mathcal{T}$.

## C  EXPERIMENTAL RESULTS

**Results on Order-II setting.** Table 5 demonstrates the comparison of SOTA PEFT methods with ours on MTIL benchmark in terms of "Transfer", "Average", and "Last" metrics (%). We label the best average results with **bold** styles.

**Learning rate and temperature of the gating mechanism.** The learning rate for the gating mechanism module is fixed at a single value across 11 tasks and the temperature during the sampling process governs the discreteness of the Gumbel logits, potentially influencing the model performance. To investigate this, we conduct ablation experiments on both learning rate and temperature of the gating mechanism, while keeping the remaining modules frozen. The results are presented in Figure 4. The results reveal that our method achieves optimal performance with a learning rate of 2.0 and a temperature of 1.0. The stability of the performance curves suggests that our approach consistently delivers high per-

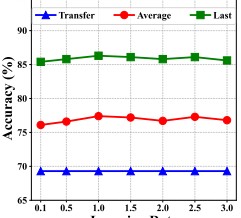
(a) Temperature is 1.0.

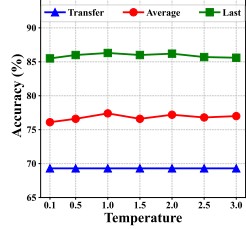
(b) Learning rate is 2.0.

Figure 4: The effects of the learning rate and temperature within the DLoRA module. We fix one and test the other. The DLoRA module exhibits robustness and insensitive to changes in two parameters.

formance across a wide range of settings, which indicates that the DLoRA module possesses a degree of robustness, remaining relatively insensitive to variations in learning rate and temperature.

**Details of experimental results on few-shot.** Table 6 demonstrates the comparison of SOTA methods with ours on 16-shot MTIL benchmark in terms of "Transfer", "Average", and "Last" metrics (%). "Ours" denotes our method. We label the best average results with **bold** styles.

**Complete results.** We present the detailed results of Order-I and Order-II in Table 7 and Table 8, which represent the classification accuracy of tasks in each incremental session.

**Visualization of individual tasks.** Figure 5 presents the distribution for 6 individual tasks. The TSP module correctly assigns task identifiers for most samples, with only a small fraction misclassified

as "unseen" (denoted by task identifier -1 in the figure) and an even smaller number incorrectly assigned to other tasks. Across all tasks, including both seen and unseen, the TSP achieves an accuracy exceeding 93%.

Table 5: Comparison with SOTA on MTIL benchmark in terms of "Transfer", "Average", and "Last" metrics (%). "Ours" denotes our method. The presented results are derived from the Order-II.

| | Method | Extra. | Param. | Aircraft | Caltech101 | CIFAR100 | DTD | EuroSAT | Flowers | Food | MNIST | OxfordPet | StanfordCars | SUN397 | *Average* |
|---|---|---|---|---|---|---|---|---|---|---|---|---|---|---|---|
| CLIP | Zero-shot | ✗ | - | 64.7 | 88.5 | 59.4 | 89.0 | 71.0 | 65.2 | 24.3 | 88.4 | 44.6 | 54.9 | 68.2 | 65.3 |
| | Full Fine-tune | ✗ | 211M | 89.6 | 92.7 | 99.6 | 94.7 | 97.5 | 81.8 | 62.0 | 95.1 | 79.5 | 98.9 | 89.6 | 89.2 |
| Transfer | ZSCL | ✗ | 211M | - | 88.3 | 57.5 | 84.7 | 68.1 | 64.8 | 21.1 | 88.2 | 45.3 | 55.2 | 68.2 | 64.1 |
| | L2P | ✗ | 0.5M | - | 70.6 | 30.7 | 78.3 | 42.8 | 38.3 | 17.4 | 75.3 | 27.4 | 23.1 | 20.7 | 42.5 |
| | DualPrompt | ✗ | 1.8M | - | 79.9 | 46.9 | 85.2 | 51.3 | 45.1 | 9.3 | 82.7 | 29.9 | 42.9 | 47.2 | 52.1 |
| | S-Prompts | ✗ | 0.5M | - | 59.8 | 46.2 | 67.7 | 47.5 | 43.8 | 13.5 | 76.8 | 31.4 | 22.6 | 43.5 | 45.3 |
| | MoE-Adapter | ✓ | 59.8M | - | 88.8 | 59.5 | 89.1 | 69.9 | 64.4 | 18.1 | 86.9 | 43.7 | 54.6 | 68.2 | 64.3 |
| | DIKI | ✗ | 1.8M | - | 85.8 | 55.3 | 89.5 | 71.1 | 62.9 | 23.7 | 93.6 | 42.1 | 43.4 | 67.9 | 63.5 |
| | Ours | ✗ | 1.8M | - | 85.7 | 64.1 | 89.1 | 70.7 | 62.6 | 24.8 | 93.3 | 43.3 | 48.4 | 68.4 | **65.0** |
| Average | ZSCL | ✗ | 211M | 81.7 | 91.3 | 91.1 | 91.0 | 82.9 | 72.5 | 33.6 | 89.7 | 53.3 | 62.8 | 69.9 | 74.5 |
| | L2P | ✗ | 0.5M | 80.1 | 87.4 | 86.7 | 89.6 | 76.8 | 59.1 | 27.7 | 79.5 | 39.9 | 34.6 | 26.5 | 62.5 |
| | DualPrompt | ✗ | 1.8M | 78.6 | 88.4 | 89.7 | 91.7 | 80.0 | 62.4 | 23.2 | 85.0 | 41.3 | 51.6 | 50.7 | 67.5 |
| | S-Prompts | ✗ | 0.5M | 79.2 | 86.5 | 89.5 | 87.0 | 78.2 | 61.5 | 25.5 | 83.6 | 41.9 | 36.3 | 47.2 | 65.1 |
| | MoE-Adapter | ✓ | 59.8M | 84.9 | 89.9 | 89.3 | 91.4 | 86.2 | 72.2 | 33.4 | 89.4 | 53.3 | 61.4 | 69.9 | 74.7 |
| | DIKI | ✗ | 1.8M | 81.8 | 89.0 | 91.3 | 93.2 | 87.8 | 70.5 | 34.0 | 94.5 | 50.9 | 53.3 | 69.6 | 74.2 |
| | Ours | ✗ | 1.8M | 82.8 | 88.7 | 93.1 | 93.0 | 87.8 | 70.5 | 36.3 | 94.4 | 52.3 | 57.4 | 70.0 | **75.1** |
| Last | ZSCL | ✗ | 211M | 78.2 | 91.1 | 97.6 | 92.5 | 87.4 | 78.2 | 45.0 | 92.3 | 72.7 | 96.2 | 86.3 | 83.4 |
| | L2P | ✗ | 0.5M | 80.1 | 89.1 | 99.1 | 93.8 | 96.2 | 76.5 | 40.1 | 86.9 | 73.5 | 86.3 | 84.2 | 82.3 |
| | DualPrompt | ✗ | 1.8M | 78.6 | 89.3 | 99.2 | 94.1 | 96.5 | 76.8 | 39.8 | 89.0 | 71.6 | 90.7 | 84.9 | 82.8 |
| | S-Prompts | ✗ | 0.5M | 79.2 | 89.1 | 99.1 | 94.3 | 95.8 | 76.3 | 39.9 | 95.5 | 70.1 | 97.6 | 84.4 | 83.8 |
| | MoE-Adapter | ✓ | 59.8M | 84.1 | 88.5 | 94.0 | 91.8 | 94.1 | 77.8 | 50.4 | 93.3 | 77.1 | 87.7 | 86.6 | 84.1 |
| | DIKI | ✗ | 1.8M | 81.8 | 89.3 | 99.3 | 94.7 | 97.4 | 76.8 | 46.4 | 96.0 | 74.2 | 98.0 | 86.0 | 85.4 |
| | Ours | ✗ | 1.8M | 82.8 | 89.0 | 99.5 | 94.5 | 97.6 | 77.0 | 50.1 | 96.3 | 76.2 | 98.0 | 85.7 | **86.1** |

Table 6: Comparison with SOTA on 16-shot MTIL-FS benchmark in terms of "Transfer", "Average", and "Last" metrics (%). "Ours" denotes our method. The presented results are derived from the Order-II.

| | Method | Aircraft | Caltech101 | CIFAR100 | DTD | Flowers | Food | StanfordCars | SUN397 | *Average* |
|---|---|---|---|---|---|---|---|---|---|---|
| CLIP | Zero-shot | 24.8 | 92.9 | 68.4 | 43.8 | 71.4 | 85.8 | 65.8 | 62.6 | 64.4 |
| | Full Fine-tune | 62.0 | 96.2 | 89.6 | 79.5 | 97.5 | 92.7 | 89.6 | 81.8 | 86.1 |
| Transfer | ZSCL | | 87.3 | 67.7 | 45.4 | 67.8 | 86.6 | 59.7 | 63.4 | 68.3 |
| | L2P | | 66.7 | 54.3 | 30.6 | 47.3 | 71.5 | 54.6 | 52.4 | 53.9 |
| | DualPrompt | | 78.8 | 64.4 | 32.0 | 51.7 | 77.5 | 49.4 | 51.3 | 57.9 |
| | S-Prompts | | 70.3 | 52.7 | 31.5 | 54.8 | 74.0 | 55.4 | 50.0 | 55.5 |
| | DIKI | | 92.7 | 68.8 | 44.1 | 70.0 | 86.2 | 65.1 | 65.5 | 70.3 |
| | Ours | | 93.3 | 68.9 | 44.4 | 70.7 | 86.5 | 66.2 | 64.8 | **70.7** |
| Average | ZSCL | 33.5 | 90.5 | 74.7 | 58.5 | 79.7 | 87.7 | 64.8 | 64.8 | 69.3 |
| | L2P | 30.2 | 84.5 | 70.1 | 51.9 | 69.6 | 77.1 | 60.0 | 55.2 | 62.3 |
| | DualPrompt | 36.5 | 89.5 | 72.5 | 52.7 | 72.3 | 80.8 | 56.1 | 54.2 | 64.3 |
| | S-Prompts | 30.6 | 86.8 | 70.0 | 51.7 | 74.3 | 78.5 | 60.7 | 53.0 | 63.2 |
| | DIKI | 41.3 | 95.3 | 76.5 | 58.5 | 82.2 | 86.4 | 68.2 | 66.6 | 71.9 |
| | Ours | 45.0 | 94.9 | 75.7 | 59.7 | 83.8 | 86.8 | 70.1 | 64.1 | **72.5** |
| Last | ZSCL | 27.7 | 90.9 | 74.4 | 64.7 | 90.2 | 89.2 | 80.6 | 74.6 | 74.0 |
| | L2P | 30.2 | 87.1 | 75.4 | 64.7 | 91.9 | 86.4 | 76.1 | 74.7 | 73.3 |
| | DualPrompt | 36.5 | 91.0 | 75.1 | 65.1 | 92.9 | 86.2 | 76.2 | 74.2 | 74.7 |
| | S-Prompts | 30.6 | 89.2 | 75.8 | 63.8 | 93.9 | 86.2 | 76.7 | 73.9 | 73.8 |
| | DIKI | 41.3 | 95.6 | 79.0 | 67.3 | 94.4 | 86.8 | 77.6 | 74.4 | 77.1 |
| | Ours | 45.0 | 95.4 | 78.3 | 68.7 | 95.7 | 87.4 | 79.4 | 75.4 | **78.2** |

**Energy and threshold.** Table 9 reveals the task identifier classification accuracy of TSP module. We evaluate the TSP module under 10 energies and 10 thresholds. The TSP module reach the optimal performance when the energy and threshold are set to 0.95 and 0.96 respectively. The results are the average accuracy for both seen and unseen tasks in all 11 incremental learning processes.

**Code.** The reproduction code is provided in "code.tar.gz" of the supplementary files.

Table 7: Accuracy (%) of our method on the MTIL benchmark with **order-I**. Each row represents the performance on every dataset of the model trained after the corresponding task. Transfer, Average, and Last metrics are shown in color.

| | Aircraft | Caltech101 | CIFAR100 | DTD | EuroSAT | Flowers | Food | MNIST | OxfordPet | StanfordCars | SUN397 | |
|---|---|---|---|---|---|---|---|---|---|---|---|---|
| **Transfer** | | 93.5 | 68.5 | 43.5 | 48.5 | 70.8 | 86.1 | 64.7 | 89.1 | 66.4 | 62.6 | 69.4 |
| Aircraft | 50.4 | 93.3 | 68.4 | 43.5 | 48.5 | 70.8 | 86.1 | 64.7 | 89.1 | 66.4 | 62.6 | |
| Caltech101 | 50.4 | 96.6 | 68.4 | 43.5 | 48.5 | 70.8 | 86.1 | 64.7 | 89.1 | 66.4 | 62.6 | |
| CIFAR100 | 50.4 | 96.6 | 86.7 | 43.5 | 48.5 | 70.8 | 86.1 | 64.7 | 89.1 | 66.4 | 62.6 | |
| DTD | 50.4 | 96.6 | 86.7 | 76.5 | 48.5 | 70.8 | 86.1 | 64.7 | 89.1 | 66.4 | 62.6 | |
| EuroSAT | 50.4 | 96.6 | 86.7 | 76.5 | 98.3 | 70.8 | 86.1 | 64.7 | 89.1 | 66.4 | 62.6 | |
| Flowers | 50.4 | 96.6 | 86.7 | 76.5 | 98.3 | 98.2 | 86.1 | 64.7 | 89.1 | 66.4 | 62.6 | |
| Food | 50.4 | 96.6 | 86.7 | 76.5 | 98.3 | 98.2 | 89.3 | 64.7 | 89.1 | 66.4 | 62.6 | |
| MNIST | 50.4 | 96.6 | 86.7 | 76.5 | 98.3 | 98.2 | 89.3 | 99.5 | 89.1 | 66.4 | 62.6 | |
| OxfordPet | 50.4 | 96.6 | 86.7 | 76.5 | 98.3 | 98.2 | 89.3 | 99.5 | 94.6 | 66.4 | 62.6 | |
| StanfordCars | 50.4 | 96.6 | 86.7 | 76.5 | 98.3 | 98.2 | 89.3 | 99.5 | 94.6 | 84.3 | 62.6 | |
| SUN397 | 50.4 | 96.6 | 86.7 | 76.5 | 98.3 | 98.2 | 89.3 | 99.5 | 94.6 | 84.3 | 77.1 | 86.5 |
| **Average** | 50.4 | 96.3 | 83.3 | 67.5 | 80.2 | 85.7 | 87.5 | 77.3 | 90.8 | 69.6 | 63.9 | 77.5 |

Table 8: Accuracy (%) of our method on the MTIL benchmark with **order-II**. Each row represents the performance on every dataset of the model trained after the corresponding task. Transfer, Average, and Last metrics are shown in color.

| | StanfordCars | Food | MNIST | OxfordPet | Flowers | SUN397 | Aircraft | Caltech101 | DTD | EuroSAT | CIFAR100 | |
|---|---|---|---|---|---|---|---|---|---|---|---|---|
| **Transfer** | | 86.1 | 64.7 | 89.1 | 70.8 | 62.6 | 24.8 | 93.3 | 43.5 | 48.5 | 68.4 | 65.2 |
| StanfordCars | 82.8 | 86.1 | 64.7 | 89.1 | 70.8 | 62.6 | 24.8 | 93.3 | 43.5 | 48.5 | 68.4 | |
| Food | 82.8 | 89.6 | 64.7 | 89.1 | 70.8 | 62.6 | 24.8 | 93.3 | 43.3 | 48.5 | 68.4 | |
| MNIST | 82.8 | 89.6 | 99.5 | 89.1 | 70.8 | 62.6 | 24.8 | 93.3 | 43.3 | 48.5 | 68.4 | |
| OxfordPet | 82.8 | 89.6 | 99.5 | 94.5 | 70.8 | 62.6 | 24.8 | 93.3 | 43.3 | 48.5 | 68.4 | |
| Flowers | 82.8 | 89.6 | 99.5 | 94.5 | 97.7 | 62.6 | 24.8 | 93.3 | 43.3 | 48.5 | 68.4 | |
| SUN397 | 82.8 | 89.6 | 99.5 | 94.5 | 97.7 | 77.5 | 24.8 | 93.3 | 43.3 | 48.5 | 68.4 | |
| Aircraft | 82.8 | 89.6 | 99.5 | 94.5 | 97.7 | 77.5 | 50.6 | 93.3 | 43.3 | 48.5 | 68.4 | |
| Caltech101 | 82.8 | 89.6 | 99.5 | 94.5 | 97.7 | 77.5 | 50.6 | 96.3 | 43.3 | 48.5 | 68.4 | |
| DTD | 82.8 | 89.6 | 99.5 | 94.5 | 97.7 | 77.5 | 50.6 | 96.3 | 76.7 | 48.5 | 68.4 | |
| EuroSAT | 82.8 | 89.6 | 99.5 | 94.5 | 97.7 | 77.5 | 50.6 | 96.3 | 76.7 | 98.2 | 68.4 | |
| CIFAR100 | 82.8 | 89.6 | 99.5 | 94.5 | 97.7 | 77.5 | 50.6 | 96.3 | 76.7 | 98.2 | 86.6 | 86.4 |
| **Average** | 82.8 | 89.3 | 93.2 | 93.0 | 87.9 | 70.7 | 36.5 | 94.4 | 52.4 | 57.5 | 70.1 | 75.3 |

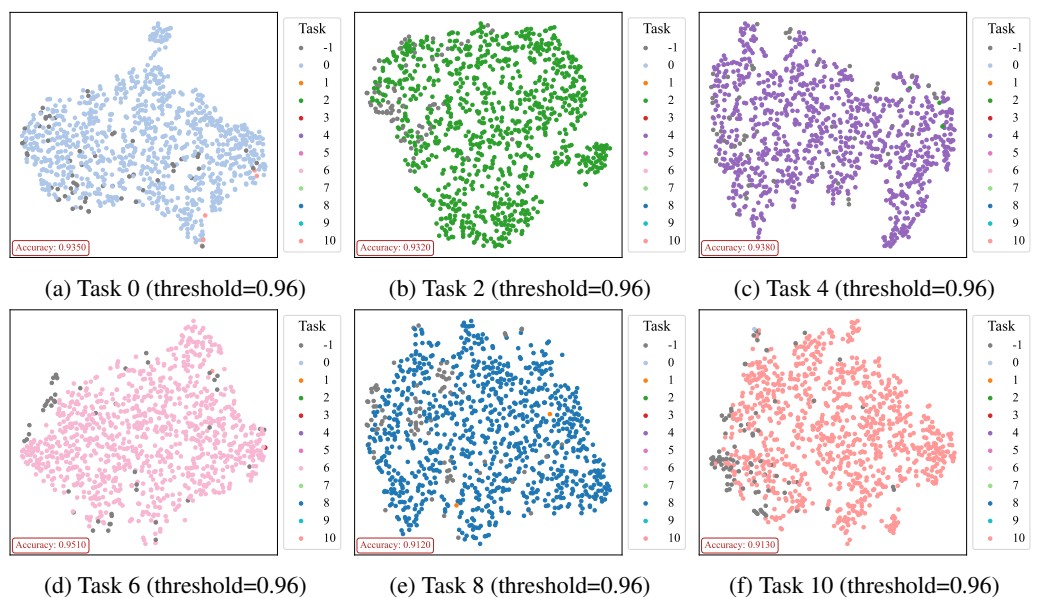

(a) Task 0 (threshold=0.96)   (b) Task 2 (threshold=0.96)   (c) Task 4 (threshold=0.96)

(d) Task 6 (threshold=0.96)   (e) Task 8 (threshold=0.96)   (f) Task 10 (threshold=0.96)

Figure 5: 6 task-specific distributions which are generated by the TSP module. We randomly select 1000 samples from each dataset. Most samples are correctly classified, as shown by their corresponding colors. A small number of samples are misclassified as from unseen tasks, while an even smaller number, though classified as from seen tasks, are incorrectly assigned to other seen tasks. Overall, the TSP module achieves an accuracy exceeding 90% for each task.

Table 9: TSP module performance under different energy and threshold setting.

| | | Threshold | | | | | | | | | |
|---|---|---|---|---|---|---|---|---|---|---|---|
| | | 0.89 | 0.90 | 0.91 | 0.92 | 0.93 | 0.94 | 0.95 | 0.96 | 0.97 | 0.98 |
| Energy | 0.89 | 0.889 | 0.912 | 0.921 | 0.909 | 0.867 | 0.800 | 0.699 | 0.589 | 0.498 | 0.462 |
| | 0.90 | 0.869 | 0.897 | 0.918 | 0.923 | 0.899 | 0.850 | 0.762 | 0.646 | 0.525 | 0.466 |
| | 0.91 | 0.846 | 0.878 | 0.908 | 0.926 | 0.920 | 0.884 | 0.811 | 0.699 | 0.559 | 0.474 |
| | 0.92 | 0.815 | 0.850 | 0.885 | 0.914 | 0.930 | 0.915 | 0.862 | 0.767 | 0.614 | 0.492 |
| | 0.93 | 0.779 | 0.815 | 0.853 | 0.888 | 0.921 | 0.930 | 0.900 | 0.827 | 0.682 | 0.520 |
| | 0.94 | 0.734 | 0.773 | 0.813 | 0.854 | 0.896 | 0.927 | 0.927 | 0.879 | 0.762 | 0.572 |
| | 0.95 | 0.682 | 0.717 | 0.758 | 0.801 | 0.847 | 0.891 | 0.928 | **0.930** | 0.844 | 0.662 |
| | 0.96 | 0.641 | 0.663 | 0.697 | 0.737 | 0.784 | 0.834 | 0.887 | 0.928 | 0.903 | 0.769 |
| | 0.97 | 0.621 | 0.628 | 0.644 | 0.670 | 0.709 | 0.758 | 0.817 | 0.879 | 0.928 | 0.874 |
| | 0.98 | 0.613 | 0.615 | 0.617 | 0.624 | 0.639 | 0.664 | 0.710 | 0.774 | 0.859 | 0.925 |

