# OpenReview forum: "Incremental Learning of Vision-Language Models via Task Subspace Projection and Dynamic LoRA"
_ICLR.cc/2026/Conference — ICLR 2026 Conference Withdrawn Submission_

### Official Review · Reviewer_v6CH · 2025-10-28

**Soundness:** 3
**Presentation:** 3
**Contribution:** 2
**Rating:** 4
**Confidence:** 3

**Summary:**

This paper proposes a novel and efficient TSP-DLoRA framework for Multi-Domain Task-Incremental Learning (MTIL). It combines a training-free task identification module (TSP) with a dynamic parameter-efficient fine-tuning module (DLoRA), striking a balance between retaining old knowledge, learning new tasks, and maintaining generalization capabilities. It achieves good performance with low parameter costs. The method's effectiveness is validated on a standard MTIL benchmark composed of 11 diverse datasets, offering a significant solution for the continual learning of large vision-language models in resource-constrained environments.

**Strengths:**

The gating mechanism in the DLoRA module allows the model to dynamically and adaptively decide whether to activate the LoRA adapters based on the features of each input sample. This is more flexible and intelligent than previous static fine-tuning methods and performs exceptionally well when dealing with multi-domain tasks that have significant distributional disparities.

**Weaknesses:**

1.	The performance of the TSP module is highly dependent on fixed energy and similarity thresholds. These values may not be optimal for all task sequences or data distributions, potentially requiring tedious manual tuning in new application scenarios.
2.	The core assumption of the TSP module is that tasks are separable in the sub feature space. For highly similar tasks with overlapping feature spaces, the TSP may struggle to create clear subspace boundaries, leading to a decrease in task identification accuracy.
3.	Although the core diagram of the paper (Figure 2) illustrates the overall architecture, the coloring of major modules is unclear. For instance, at the bottom of the training flowchart, the colored feature blocks representing different tasks (e.g., aircraft, food, flowers) lack clear labels or explanations. This makes it difficult for readers to fully understand their specific roles in the 'task subspace' construction process, hindering detailed comprehension and reproducibility.

**Questions:**

see weaknesses.

---

### Official Review · Reviewer_nNiC · 2025-10-29

**Soundness:** 2
**Presentation:** 3
**Contribution:** 2
**Rating:** 4
**Confidence:** 5

**Summary:**

This paper proposes a novel approach for multi-domain task-incremental learning (MTIL) in vision-language models, termed TSP-DLoRA.
The method integrates two main components, Task Subspace Projection (TSP) and Dynamic Low-Rank Adapter (DLoRA). TSA is a training-free discriminative module that identifies task boundaries by projecting sample features into task-specific subspaces. It distinguishes seen versus unseen tasks via cosine similarity and an adaptive threshold, thereby mitigating task confusion and catastrophic forgetting. DLoRA is an adaptive parameter-efficient fine-tuning module that dynamically determines where to apply LoRA based on task complexity, guided by a Gumbel-based gating mechanism.
Experiments are conducted on 11 datasets from the MTIL benchmark (e.g., CIFAR100, EuroSAT, Flowers, Food, SUN397), showing state-of-the-art performance across “Transfer”, “Average”, and “Last” metrics, while training only 0.86% of parameters. The method also reduces GPU cost and inference time compared with MoE-Adapter and DIKI.

**Strengths:**

1.Achieves SOTA results while training <1% of parameters. It’s a remarkable trade-off between performance and efficiency.

2.The combination of SVD-based task subspace analysis with adaptive LoRA gating is novel and creative.

3.The experiment is complete, which covers 11 datasets, strong baselines, and ablation studies that isolate both modules’ effects.

**Weaknesses:**

1.While the modular ablations are solid, there’s limited discussion of failure cases

2.The paper could better justify why cosine similarity with a fixed threshold generalizes across domains.

3.Would fine-tuning the CLIP backbone further improve unseen-task performance?

4.Have you considered extending TSP-DLoRA to cross-modal task transfer (e.g., from image-text to video-text)?

**Questions:**

see weaknesses

---

### Official Review · Reviewer_5Get · 2025-11-01

**Soundness:** 2
**Presentation:** 2
**Contribution:** 1
**Rating:** 2
**Confidence:** 5

**Summary:**

The paper proposes a Cross-Task Identification (CTI) module that predicts the task/dataset identity of an input image and routes it to a corresponding dataset-specific LoRA (DLoRA) adapter on top of a CLIP-style image encoder. The goal is to improve adaptation to new domains while retaining zero-shot generalization. Experiments are conducted under the MTIL benchmark.

**Strengths:**

1. The CTI→DLoRA routing pipeline is a clean, extensible design for multi-domain adaptation.

2. Routing prior to specialized adapters is a reasonable way to reduce negative transfer when tasks are heterogeneous.

**Weaknesses:**

1. **The paper’s CTI-centric design conflicts with the MTIL evaluation protocol**: under MTIL, each dataset is evaluated in isolation using only its own label set, which assumes the test sample’s dataset/task is known a priori. In this setting, a task-identification module adds no value—one can directly route to the corresponding dataset-specific DLoRA.

2. **Limited contribution.** The architecture is nearly isomorphic to MoE-adapter , i.e., a task identifier plus task-specific LoRA. Even Figure 2 appears very similar to MoE-adapter schematics, making the contribution look incremental.

3. **Marginal performance gains.** On MTIL, the reported Transfer (Avg Last) improvements over MOE-ADAPTER are small, calling into question the practical benefit relative to added complexity.


4. **Insufficient baselines.** For example:
ConDU: Enhanced Continual Learning of Vision-Language Models with Model Fusion
RAIL: Advancing Cross-domain Discriminability in Continual Learning of Vision-Language Models
DuPLe: Mitigating the Evolving Semantic Entanglement in Continual Learning of Vision-Language Models

**Questions:**

**Transfer > Zero-shot on unseen tasks?**

You state that for unseen tasks the system falls back to the original CLIP (even assuming a perfectly accurate task identifier offers no specialized adapter). How do you obtain transfer scores higher than zero-shot on some datasets (e.g., 93.5 vs. 88.4 on Caltech101)?

---

### Official Review · Reviewer_bK4Y · 2025-11-01

**Soundness:** 2
**Presentation:** 2
**Contribution:** 1
**Rating:** 2
**Confidence:** 4

**Summary:**

The paper proposes a Cross-Task Identification (CTI) module that predicts the task or dataset identity of an input image and routes it to a corresponding dataset-specific LoRA (DLoRA) adapter built on top of a CLIP-style image encoder. The goal is to enhance adaptation to new domains while preserving zero-shot generalization capability. Experiments are conducted on the MTIL benchmark to evaluate the effectiveness of the proposed approach.

**Strengths:**

The CTI→DLoRA routing pipeline presents a clean and extensible design for multi-domain adaptation.
Performing routing prior to specialized adapters is a reasonable and effective strategy to mitigate negative transfer across heterogeneous tasks.

**Weaknesses:**

[Major]1-Novelty is incremental: The proposed architecture is largely isomorphic to MoE-Adapter, comprising a task identifier coupled with task-specific LoRA modules. Even Figure 2 closely resembles the schematic of MoE-Adapter, making the overall contribution appear incremental.

[Major]2-Limited performance gains: On MTIL, the reported Transfer (Avg Last) improvements over MoE-Adapter are marginal, raising questions about the practical benefits relative to the added architectural complexity.

[Major]3-Unfair comparison: Miss several methods, for example:
ConDU: ENHANCED CONTINUAL LEARNING OF VISION-LANGUAGE MODELS WITH MODEL FUSION
RAIL: Advancing Cross-domain Discriminability in Continual Learning of Vision-Language Models
DuPLe: Mitigating the Evolving Semantic Entanglement in Continual Learning of Vision-Language Models

**Questions:**

Please see weakness.

---

### Note · Authors · 2025-11-12

I have read and agree with the venue's withdrawal policy on behalf of myself and my co-authors.